# Effect of SCUBA Diving on Ophthalmic Parameters

**DOI:** 10.3390/medicina58030408

**Published:** 2022-03-09

**Authors:** Laurent Deleu, Janet Catherine, Laurence Postelmans, Costantino Balestra

**Affiliations:** 1Cliniques Universitaires de Bruxelles, Department of Ophthalmology, Hôpital Erasme, Université Libre de Bruxelles (ULB), Route de Lennik 808, 1070 Brussels, Belgium; 2CHU Brugmann, Department of Ophthalmology, Université Libre de Bruxelles (ULB), Site Victor Horta Place Arthur Van Gehuchten 4, 1020 Brussels, Belgium; janet.catherine@gmail.com (J.C.); laurencedominique.postelmans@chu-brugmann.be (L.P.); 3Laboratory of Environmental and Occupational (Integrative) Physiology, Haute Ecole Bruxelles-Brabant, 1160 Brussels, Belgium; 4Physical Activity Teaching Unit, Motor Sciences Department, Université Libre de Bruxelles (ULB), 1050 Brussels, Belgium

**Keywords:** subfoveal choroidal thickness, intraocular pressure, central corneal thickness, central serous chorioretinopathy, flow-mediated dilation, arterial stiffness, endothelial dysfunction

## Abstract

*Background and Objective*: Several cases of central serous chorioretinopathy (CSC) in divers have been reported in our medical retina center over the past few years. This study was designed to evaluate possible changes induced by SCUBA diving in ophthalmic parameters and especially subfoveal choroidal thickness (SFCT), since the choroid seems to play a crucial role in physiopathology of CSC. *Materials and Methods*: Intraocular pressure (IOP), SFCT, pachymetry, flow-mediated dilation (FMD), blood pressure, and heart rate were measured in 15 healthy volunteer divers before diving, 30 and 60 min after a standard deep dive of 25 m depth for 25 min in a dedicated diving pool (NEMO 33). *Results*: SFCT reduces significantly to 96.63 ± 13.89% of pre-dive values (*p* = 0.016) 30 min after diving. It recovers after 60 min reaching control values. IOP decreases to 88.05 ± 10.04% of pre-dive value at 30 min, then increases to 91.42 ± 10.35% of its pre-dive value (both *p* < 0.0001). Pachymetry shows a slight variation, but is significantly increased to 101.63 ± 1.01% (*p* = 0.0159) of the pre-dive value, and returns to control level after 60 min. FMD pre-dive was 107 ± 6.7% (*p* < 0.0001), but post-dive showed a diminished increase to 103 ± 6.5% (*p* = 0.0132). The pre-post difference was significant (*p* = 0.03). *Conclusion*: Endothelial dysfunction leading to arterial stiffness after diving may explain the reduced SFCT observed, but SCUBA diving seems to have miscellaneous consequences on eye parameters. Despite this clear influence on SFCT, no clear relationship between CSC and SCUBA diving can be drawn.

## 1. Introduction

Diving is a recreational and professional activity with many potential risks. Effects of SCUBA diving on the eye have been often reported, including retinal complications (ocular barotrauma, decompression sickness syndrome, arterial gas embolism, ultraviolet keratitis, choroidal ischemia, retinal vein occlusion, central serous chorioretinopathy, variation in color/contrast sensibility, etc.) [1,2,3,4,5].

Central serous chorioretinopathy (CSC) is a potentially severe ocular disease of the retina characterized by recurrent and/or persistent subretinal fluid, causing severe retinal pigment epithelial alterations and a variable degree of visual loss. In our medical retina center, we noticed the presence of common history of SCUBA diving among CSC patients representing around 4% of cases during a two-year period. It is known that CSC is commonly associated with choroidal hyperpermeability and increased subfoveal choroidal thickness (SFCT), but its physiopathology is still poorly understood [6,7].

Considering the importance of SFCT in the diagnostic of CSC, we studied the effect of diving on SFCT and other parameters before and after a deep dive.

## 2. Methods

### 2.1. Patients and Timing of the Study

For this study, 15 volunteer healthy divers were first interrogated about their medical and diving history, and signed an inform consent. Divers were otherwise healthy Caucasian males between 28 and 72 years old (median 48.93 years). Body mass index 20–25, good general health, nonsmoking (except one), and certified as “advanced divers” with at least 50 logged dives. Of these, three presented treated arterial hypertension, and one with diabetes mellitus without related ophthalmic impairment. Exclusion criteria included previous eye diseases. The study protocol was approved by the Local Ethic Committee Brussels (Academic Bioethical Committee, Brussels, Belgium. Reference Number: B200-2020-088. Date: 10 October 2020), and each subject gave written informed consent before participation. All studies were performed in accordance with the Declaration of Helsinki [8]. The study is part of a series of ongoing field studies on vascular gas emboli (VGE).

All divers received one drop of tropicamide 0.5% (5 mg/mL) in both eyes. Then, 20 min after instillation, they were examined for intraocular pressure (IOP), pachymetry (PACHY), keratometry (KR), subfoveal choroidal thickness (SFCT), and retinal autofluorescence. They were all tested for post-ischemic flow-mediated dilation (FMD), blood pressure (BP), and heart rate (HR). They also underwent transthoracic heart echography (TTE). After pre-dive measures, all divers went for 25 m dive for 25 min in Nemo 33 diving pool in Brussels. Half of the divers underwent measures during the morning, and the other half in the afternoon. All divers respected published decompression tables while coming back to the surface.

Respectively 30 and 60 min after regaining surface, the patients were retested for the same ocular tests. FMD of brachial artery was measured 30 min after regaining surface. BP, HR, and TTE were also controlled after diving looking for post-dive vascular gas emboli (VGE). All divers drank 500 mL water 60 min after the dive.

### 2.2. Data Acquisition

#### 2.2.1. Ophthalmic Measurements

Nidek Tonoref III (NIDEK Co., Ltd., Tokyo, Japan) was used to measure IOP, PACHY, and KR. Pachymetry is here defined by the central corneal thickness, and keratometry is defined by the corneal curvature of the main corneal meridians. Results were calculated using mean of three values for each eye. IOP was not corrected by pachymetry. Nidek Tonoref III provides accurate and reliable measurements of IOP, compared with Goldmann applanation tonometry, considered as the gold-standard for IOP measurements [9]. Even older devices of air tonometers show low variability ranging from −2 mmHg to +2 mmHg, principally due to the cardiac cycle, explaining why three measures should be taken in regular practice [10].

SFCT was measured with enhanced depth imaging (EDI) modality using a Spectral Domain-Optical Coherence Tomography (SD-OCT) (Spectralis, wavelength: 870 nm; Heidelberg Engineering Co. Manufacturer: Heidelberg Engineering GmbH, Heidelberg, Germany). The same device was used for retinal autofluorescence imaging. SD-OCT gives two or three-dimensional images of the retina with near-cellular resolution, allowing ophthalmologists to analyze histologic-like images. SD-OCT uses near-infrared wavelength, and thus does not expose patients to radiation. The procedure for SFCT measurement that we used was previously described by Spaide et al. [11], and is defined as the vertical distance from the hyperreflective line of Bruch’s membrane to the hyperreflective line of the inner surface of the sclera. All images were taken by one clinician, and were assessed by three different clinicians. Clinicians were blinded, as they did not know the patient’s identification and timing of the measure (pre or post dive). Personal keratometry was encoded in Heidelberg OCT to improve each diver’s measurements.

#### 2.2.2. Flow-Mediated Dilation (FMD)

FMD, an established measure of the endothelium-dependent vasodilation mediated by nitric oxide (NO) [12], was used to assess the effect of diving on main conduit arteries. Subjects were at rest for 15-min in a supine position before the measurements were taken. They were asked not to drink caffeinated beverages for the 6 h preceding measurements. Subjects were instructed not to perform strenuous physical exercise 24 h before, or stay in altitude up to 2 weeks before and during the entire study protocol. Brachial artery diameter was measured by means of a 5.0–10.0 MHz linear transducer using a Mindray DP-30 digital diagnostic ultrasound system immediately before and 1-min after a 5-min ischemia induced by inflating a cuff placed on the forearm to 180 mmHg as previously described [13].

All ultrasound assessments were performed by an experienced operator, with more than 100 scans/year, which is recommended to maintain competency with the FMD method [14].

When the images were chosen for analysis, the boundaries for diameter measurement were identified manually with an electronic caliper (provided by the ultrasonography software) in a threefold repetition pattern to calculate the mean value. In our laboratory, the mean intra-observer variability for FMD measurement for the operator (CB) recorded the same day, on the same site, and on the same subject was 1.2 ± 0.2%.

Post-dive values were obtained 20–30 min after surfacing. The divers were given a specific time to enter into the water with their companion (buddy) in order to make it possible to respect the tight timing after the dive for the measurements to be taken. FMD were calculated as the percent increase in arterial diameter from the resting state to maximal dilation.

#### 2.2.3. Post Diving Vascular Gas Emboli (VGE)

Post-dive vascular gas emboli (VGE) (decompression bubbles) were observed using transthoracic echocardiography and a “frame-based” counting method for VGE recently described [15], allowing for continuous values and parametric statistical approaches. Echocardiography was done with a Vivid-I portable echocardiograph (Manufacturer: GE Healthcare, Chalfont Saint Giles, UK) used at the poolside; echocardiography loops were recorded on hard disk for offline analysis by three blinded evaluators. VGE numbers were counted at 30 min and 60 min post dive.

## 3. Statistical Analysis

The normality of data was performed by means of Shapiro–Wilk or D’Agostino-Pearson tests. When a Gaussian distribution was assumed, they were analyzed with a one-way ANOVA for repeated measures with Dunnett’s post-hoc test; when comparisons were limited to two samples, paired or non-paired *t*-test were applied. If the Gaussian distribution was not assumed, the analysis was performed by means of a non-parametric multiple comparisons using Dunn’s test or, if limited to two samples, a Wilcoxon test. Taking the baseline measures as 100%, percentage changes were calculated for each diver, allowing for an appreciation of the magnitude of change rather than the absolute values. All statistical tests were performed using a standard computer statistical package, GraphPad Prism version 5.00 for Windows (GraphPad Software, San Diego, CA, USA). A threshold of *p* < 0.05 was considered statistically significant. All data are presented as mean ± standard deviation (SD). Sample size was calculated setting the power of the study at 95%, and assuming that variables associated to diving would have been affected on a similar extent than that observed in our previous studies [16,17,18].

## 4. Results

### 4.1. Generalities

All results are expressed in a percentage of pre-dive values (relative values) rather than absolute values. Studied parameters and especially SFCT have high interindividual variability. As each diver acts as his/her own control, results were expressed in percentage of pre-dive values so that this proportion could be compared and analyzed with others. As explained in statistical analysis section, taking the baseline measures as 100%, percentage changes were calculated for each diver, allowing an appreciation of the magnitude of change rather than the absolute values.

### 4.2. Diving Related

FMD comparison between pre/post dive situation and control values is shown in Figure 1. FMD in our divers is increased in pre-dive situation (107.15 ± 6.6% (*p* < 0.0001)). This dilatation is significantly reduced after the dive (103 ± 6.5%, *p* = 0.026). This reduction between the two conditions shows a difference consistent with previously published data and is significant (*p* = 0.027) [19].

All divers underwent trans-thoracic echocardiography (TTE) 30 min and 60 min after the dive. Figure 2 shows that divers had, on average, eight bubbles per heartbeat (8.0 ± 9.3; mean ± SD) in the right heart (ventricle or atrium) 30 min after the dive. This number is reduced to a mean of 5.25 ± 8.8 Bubbles per HB. The number of circulating decompression bubbles per heartbeat after an hour post the dive was significantly reduced. No participant had symptoms of decompression disease, and the number of VGE found and its decrease is consistent with previously published data [20,21].

### 4.3. Ophthalmological

Comparisons between pre and post-dive values appear in Table 1, and are expressed as a percentage relative to pre-dive values. SFCT and IOP both decrease significantly 30 min after the dive. SFCT recovers pre-dive thickness after 60 min, but IOP is still diminished after an hour. Corneal thickness tends to slightly increase, but the change is no longer significant after 60 min. Table 2 presents mean ± SD of measurements. Ophthalmic data post-dive are significantly altered 30 min post-dive, most of which returned to basal levels 60 min post diving, except IOP, which stayed still reduced 60 min post dive. Figure 3 shows OCT imaging of normal posterior pole of the eye, with color code to distinguish retina from choroid and sclera. This figure helps better understanding of Figure 4, which shows comparison between SFCT before and 30 min after diving in a same eye. High magnification was used to highlight the 11 µm difference in this case.

## 5. Discussion

### 5.1. Subfoveal Choroidal Thickness

The choroid is a vascular layer situated between the sclera and the retina. It is composed of several layers: the choriocapillaris, 10 µm-thick capillary network; the Sattler’s layer, composed of arterioles, small arteries, and veins; Haller’s layer, composed of larger blood vessels; the suprachoroid, which is non-vascular, composed of melanocytes, fibroblasts and collagen; and the lamina fusca, separating the choroid from the sclera [22]. It is a highly vascularized space, as the flow per perfused volume is the highest of any other human tissue [23]. While it is well-known that myopic eyes (with greater axial length) tend to have thinner choroid, myopic shift can induce thickening of SFCT in animals, while hyperopic shift induces thinning of SFCT [24]. Mechanisms remain hypothetic, but Wallman et al. observed that, at least in birds, SFCT variation is linked to expansion or compression of lacunae present in the outer choroid [25]. Liquid redistribution seems to be the key of SFCT variation by different suggested mechanisms [24]. However, people with myopic eyes tend to have thinner choroid than emmetropic or hyperopic eyes. These factors make it difficult to know what abnormal SFCT is or not. Studies may give different normal values. Akhtar et al. concluded that subfoveal choroidal thickness was 307 ± 79 µm in an Indian population of any age [26]. Entezari et al. concluded 363 ± 84 µm in an Iranian population [27]. Karapetyan et al. compared SFCT in Caucasians, Africans, and Asians populations, and concluded no significant differences between those ethnics. The mean SFCT in Caucasians was 403.62 ± 37.4 µm. The literature presents sometimes normal SFCT ranges that may be different from ours [28]. Moreover, SFCT seems to decrease over day time [11]. All of these considerations show the importance of measuring variations taking the baseline measures as 100%, for each diver, allowing an appreciation of the magnitude of change rather than the absolute values.

To the best of our knowledge, acute effects of diving on SFCT have never been studied. Our study shows transient and significant SFCT decrease in the 30 first minutes following a deep dive. After 60 min, SFCT returns to its initial thickness. Different hypotheses were investigated to explain those results.

Our preferred hypothesis is that SFCT decreases due to vascular phenomena. As previously said and confirmed by other studies, FMD decreases after a deep dive due to endothelial dysfunction [29,30]. Considering the density of blood vessels in the choroid, reduced dilation of this vascular meshwork could explain by itself the reduction in SFCT. Moreover, diving involves an increase in oxygen partial pressure leading to oxidative stress by increased free radical concentration in blood. This contributes to endothelial dysfunction and, ultimately, arterial stiffness. It seems likely that vascular smooth muscles also have a role in FMD reduction, but various results are found in studies [29,31]. Implications of smooth-muscle cells might be important, as it has been shown that non-vascular smooth-muscle cells are present in primates in the suprachoroid (just next to lacunae), and in a single layer just beneath Bruch’s membrane [32]. It has been suggested that contraction of those smooth-muscle cells oppose the tendency of lacunae to gain fluid [24]. Insufficient relaxation of these cells might prevent swelling of the lacunae, inducing vascular shift from the choroid to general circulation, decreasing SFCT. Thus, both arterial stiffness due to endothelial dysfunction and insufficient relaxation in choroidal smooth-muscle cells reduce the choroidal intravascular fluid and are an explanation to reduced SFCT post diving.

In the literature, SFCT tends to be negatively correlated to IOP [33,34,35,36,37,38]. As both IOP and SFCT decrease in our study, IOP variation does not explain the SFCT decreases. It is also interesting to notice that pachymetry increases in our study and would be expected to falsely elevate the IOP. We can confidently conclude that changes in IOP and pachymetry are not responsible for the diminution of SFCT after the dive. It is a good argument to suggest that there is an extra-ophthalmological phenomenon explaining SFCT reduction.

Other hypotheses were also considered as being unlikely with our experimental setting:

Effects of physical exercise on SFCT have been reported but results are various and not well established [39,40,41].

Tropicamide was instilled before pre-dive measures, therefore there was no influence in comparison between pre- and post-dive values. Iovino et al. published in 2020 that there was no significant difference in SFCT before or after mydriatic instillation [42].

SFCT presents diurnal variation and tends to decrease over the day. Some divers underwent measures in the morning, other in the afternoon. Both are periods of decreasing SFCT, so we do not expect any difference between those two cohorts for relative measurements. Moreover, this circadian cycle does not explain the re-increase of SFCT 60 min after the dive. This points out an extra-physiological phenomenon allowing us to exclude this hypothesis.

### 5.2. Flow-Mediated Dilation (FMD) and Vascular Gas Emboli

A nitric oxide (NO) mediated change in the surface properties of the vascular endothelium favoring the elimination of gas micronuclei has previously been suggested to explain this protection against bubble formation [43]. It was shown that NO synthase activity increases following 45 min of exercise, and that NO administration immediately before a dive reduces VGE [44]. Nevertheless, bubble production is increased by NO blockade in sedentary but not in exercised rats [45], suggesting other biochemical pathways such HSPs, antioxidant defenses or blood rheology.

Vascular gas emboli are probably involved in the post dive reduction of FMD. Nevertheless the available literature refrains us to draw a direct link between FMD reduction and VGE, since micro and macro vascularization react differently [30], and different preconditioning procedures before diving have specific actions independently on FMD and VGE, while others interfere with both [20].

It appears that FMD seems more linked to oxygen partial pressure changes during diving, whereas VGE are more depending on preexisting gas micronuclei population in the tissues and vascular system and coping with inflammatory responses [20,46,47].

FMD is a marker of endothelial function and is reduced in the brachial artery of healthy divers after single or repetitive dives [48,49]. This effect does not seem to be related to the amount of VGE, and was partially reversed by acute and long-term pre-dive supplementation of antioxidants, implicating oxidative stress as an important contributor to post-dive endothelial dysfunction [29]. Decreased nitroglycerin-mediated dilation after diving highlights dysfunction in vascular smooth-muscle cells as possible etiology of those results [29]. Very recent data show that the FMD reduction encountered after a single dive without presence of VGE, is comparable to the reduction encountered with the presence of VGE [19].

Our results about FMD are in tune with what has been previously described in literature on the subject and, with its consequences, is the most likely explanation to decreased SFCT observed in this study.

### 5.3. Central Serous Chorioretinopathy

CSC is characterized by localized serous retinal detachment associated with focal altered retinal pigment epithelium. Known risk factors are genetics, male gender, cardiovascular diseases and arterial hypertension, increased corticosteroids blood concentration by any income, pregnancy, psychopathology (type A personality), peptic ulcer and Helicobacter Pylori, some drugs (including phosphodiesterase-5 inhibitors), and sleep disturbances. Despite being a common chorioretinal disease, pathophysiology of CSC remains ambiguous. Advances in imaging techniques have shown that CSC is associated with localized areas of delayed choriocapillaris perfusion, congestion of choroidal vessels, choroidal hyperpermeability, and increased SFCT, causing damage to the retinal pigmented epithelium. Imbalance in mineralocorticoid pathway has also been suggested as potential cause. Sympathetic overaction and a decreased parasympathetic tone might also play a role [7,50].

In our department of ophthalmology, there have been several cases of CSC in SCUBA divers in the past years. However, CSC in divers was rarely described in literature [4]. The relationship between hyperbaric environment and CSC could be easily overlooked.

A hypobaric environment might also have influence on SFCT. CSC was reported in at least four air pilots [51,52,53,54] and a case during hypobaric chamber exposure [55] together with a small but significant CCT increase was described in high-altitude exposure [56].

Diving has been suspected to cause macular damages since 1988 by the study of Polkinghorne et al. [57]. They highlighted that divers had significantly more retinal pigment epithelial defects, and the prevalence of defects increased with years of diving experience and history of decompression sickness. However, many other studies revealed no significant differences with control groups regarding retinal pigment epithelial alterations [58,59,60]. A total of three eyes (10%) were found to present retinal epithelium alterations in our study. It is difficult to know if those are the results of diving practice or if it is just incidental finding similar to general population. Decrease in SFCT has been demonstrated in our study, while CSC is typically described with the pachychoroid, which is the opposite.

Regarding those elements, it seems uncertain if SCUBA diving is a risk factor for CSC. Transient decreased SFCT would not explain increased CSC incidence. We did not observe more macular damage, significant retinal pigment alterations nor increased SFCT in our divers compared with general population seen in our daily practice. It seems more likely that cases of CSC in divers reported in our center are just coincidental, as patients had other risk factors of CSC.

### 5.4. Intraocular Pressure

Our results demonstrated decrease of 88.05% (*p* < 0.0001%) in IOP 30 min after diving.

Lowered IOP is widely described following physical exercise. However, it still remains a poorly understood phenomenon [61]. A total of three theories of its etiology involve decreased blood pH, elevated blood plasma osmolarity, and elevated blood lactate [62]. Increases in trabecular meshwork thickness, area, and perimeter of Schlemm’s canal have also been observed after physical activity, and are thought to be a consequence of sympathetic response to exercise. It was not significantly correlated with the decrease in IOP [63]. Similar observations are expected in SCUBA diving, and it is the most likely hypothesis to explain our results. Other hypotheses were also considered as being unlikely with our experimental setting:

Goenadi et al. suggest that in contrast with swimming goggles, diving masks can induce small decrease of 0.43 mmHg in IOP after diving [64]. All divers wore diving masks (different from swimming goggles), respected mask pressure normalization during diving, and no mask squeeze was observed.

Corneal parameters as central corneal thickness and external curvature radius have influence on IOP measurements [65]. Increased pachymetry is associated with overestimated IOP measures, which also does not explain the results. Intraocular bubbles might block trabecular outflow, increasing IOP.

Fadini et al. interestingly showed that patients without any cardiovascular risk factors but suffering from ocular hypertension and primary open-angle glaucoma (POAG) had both FMD and endothelial progenitor cell (EPC) reduced [66]. It seems likely that chronic reduced FMD and endothelial dysfunction increase IOP. However, it was not demonstrated in transient FMD variation. No other description of decreased IOP after SCUBA diving was found on Pubmed. In contrast to our results, Maverick et al. showed increase IOP post-dive negatively correlated to pachymetry [67].

Instillation of tropicamide was made before pre-dive measures, and so does not explain our results. Effects of tropicamide on IOP vary in literature [68,69,70,71,72].

Our results on IOP are in tune with previous papers studying influence of sports on IOP [61]. Also, as IOP is negatively correlated to SFCT, we can formally exclude the role of IOP in the diminution of SFCT after the dive. It is a good argument to suggest that there is an extra-ophthalmological phenomenon explaining SFCT reduction.

### 5.5. Pachymetry

This study shows increase in pachymetry 30 min after diving (101.6 ± 1.0%; *p* = 0.015). Results are not significant anymore at 60 min.

Maverick et al. described no significant change of pachymetry in 24 eyes after diving from 34 to 100 feet of depth [67]. However, in a major review, Butler et al. explained how the use of diving mask may, if the divers do compensate air compression in the mask exhaling gas through the nose into the mask, cause a negative pressure around the eye [1]. In severe cases, this can lead to ocular barotrauma. We can easily imagine that this negative pressure may be responsible for increased pachymetry after the dive.

Increased pachymetry also does not explain the SCFT decrease nor IOP decrease.

## 6. Conclusions

SCUBA diving appears to have miscellaneous consequences on ophthalmic parameters. We postulate that SFCT is transiently reduced as a consequence of vascular changes, involving increased arterial stiffness and insufficient relaxation in vascular smooth-cells due to oxidative stress and endothelial dysfunction. IOP showed transient decrease until 60 min after the dive, and was not correlated with changes of SFCT or pachymetry. It is a strong argument to point out an extra-ophthalmic phenomenon to explain our results. The results brought us no argument to conclude in a relationship between CSC and SCUBA diving.

## Figures and Tables

**Figure 1 medicina-58-00408-f001:**
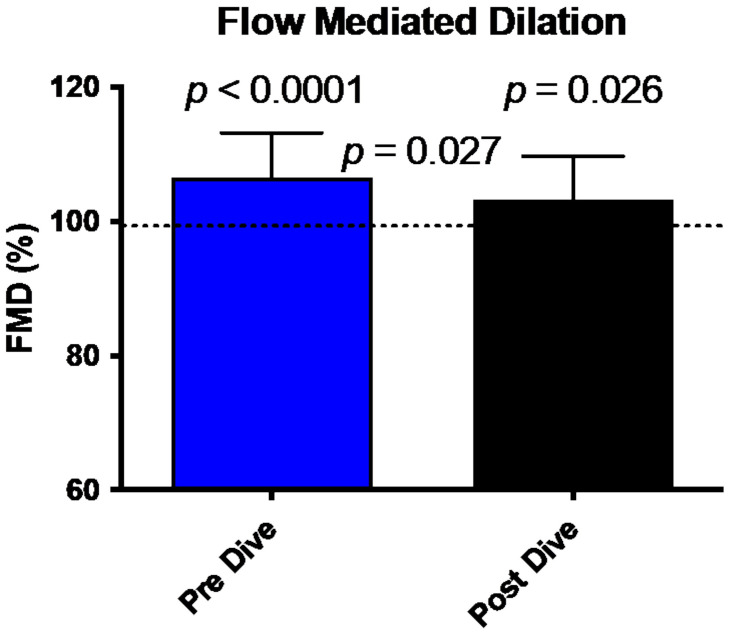
Bar graph illustrating the reduction of FMD 30 min post dive (black bar) compared to predive values (Blue Bar) (mean ± SD). (*N* = 15).

**Figure 2 medicina-58-00408-f002:**
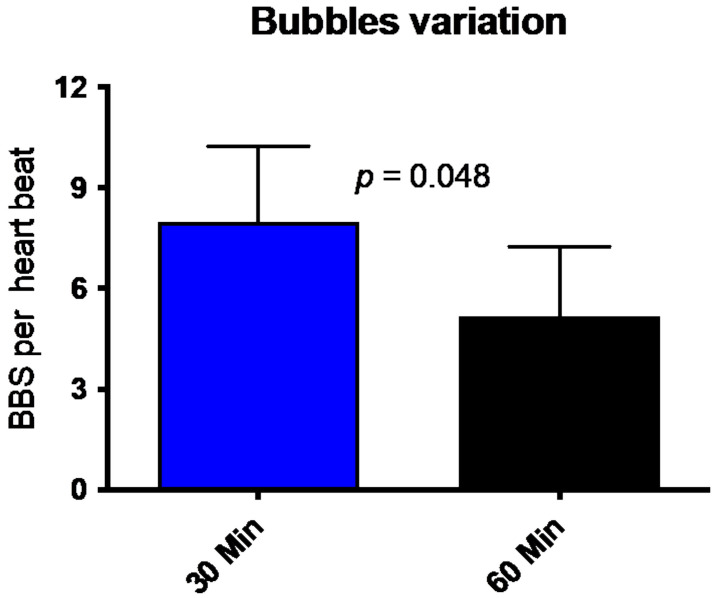
Number of bubbles (BBS = Bubbles) per heartbeat 30 min (blue bar) and 60 min (black bar) after diving. (Mean ± SD).

**Figure 3 medicina-58-00408-f003:**
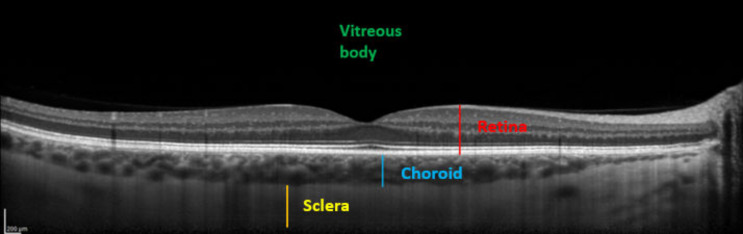
Enhanced depth imaging-optical coherence tomography (EDI-OCT) showing different structures. From anterior to posterior: Vitreous body (green), neurosensory retina and the retinal pigment epithelium (red), choroid (SFCT in blue), and sclera (yellow).

**Figure 4 medicina-58-00408-f004:**
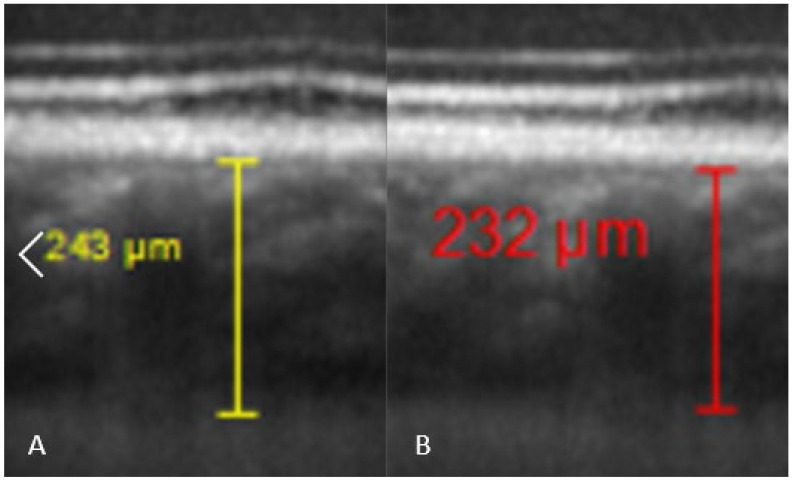
Comparison of SFCT in the same diver before the dive ((**A**), SFCT estimated at 243 µm by one of the investigators), and 30 min after the dive ((**B**), SFCT estimated at 232 µm).

**Table 1 medicina-58-00408-t001:** Comparison (in percentage of pre-dive values) of SFCT, IOP, and pachymetry between pre- and post-dive values at 30 min and 60 min post dive.

Mean ± SD	Pre-Dive	30 min Post-Dive	*p*-Value	60 min Post-Dive	*p*-Value
SFCT (%)	100	96.6 (±13.89)	0.016	98.4 (±5.7)	0.21 (ns)
IOP (%)	100	88.05 (±10.03)	<0.0001	91.4 (±10.3)	<0.0001
Pachymetry (%)	100	101.6 (±1.0)	0.015	100.2 (±1.4)	ns

(ns = not significant).

**Table 2 medicina-58-00408-t002:** Mean measures ± SD of SFCT (expressed in µm), IOP (expressed in mmHg) and pachymetry (expressed in µm).

Mean ± SD	Pre-Dive	30 min Post-Dive	*p*-Value	60 min Post-Dive	*p*-Value
SFCT (µm)	327.1 (±102.0)	318.1 (±109.7)	0.0326	322.2 (±102.1)	0.2434
IOP (mmHg)	16.4 (±2.009)	14.3 (±2.27)	<0.0001	14.98 (±2.67)	<0.0001
Pachymetry (µm)	559.3 (±27.83)	566.5 (±33.53)	0.0120	562.9 (±28.62)	ns

(ns = not significant).

## Data Availability

Data are available at request from the authors.

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
