# Peer review of "Effect of SCUBA Diving on Ophthalmic Parameters"

_medicina, 2022, doi:10.3390/medicina58030408_

Round 1

Reviewer 1 Report

Dear authors,

Thank you for submitting this work for publication. This is an interesting study about some of the possible effects that diving can cause on the physiology of the cornea, IOP, and retina.

Overall the work is well designed and written, I found the exposure of the methods a bit confusing and could be simplified for the non-experts readers (i.o. explaining briefly with is an OCT or pachymetry). 

You mentioned that IOP was measured using the Nidek Tonoref III, could you explain what is the assumed normal difference between measurements that is due to the variability of the device? because you found a significant change in IOP that I assume is considering that the variability between measures is 0, which probably is not true. Also, you say that the change in the IOP is 88% which considering that the initial mean is 16.35 and the 30 minutes post is 14.30 I am not sure if its right, I would say that a decrease of 2 mmHg if 16.35 is 100% represents 12.2% of decrease, would you agree on that? 

In the 5.4. Intraocular Pressure point, you mention that no intraocular bubbles were identified in the eyes of the divers, how did you assess that? did you perform gonioscopy to assess properly the angle, or use anterior segment OCT? otherwise, it seems difficult to affirm that microscopic bubbles were not present at all, and that possibility might be highlighted or trying to seek another explanation.

Why you did not consider adjusting the IOP with the corneal thickness? it is a shame as you had all the data collected to do it and it would add more accuracy to your results.

Author Response

(I also linked an attachment if needed)

Thank you for submitting this work for publication. This is an interesting study about some of the possible effects that diving can cause on the physiology of the cornea, IOP, and retina.

1/ Overall the work is well designed and written, I found the exposure of the methods a bit confusing and could be simplified for the non-experts readers (i.o. explaining briefly with is an OCT or pachymetry). 

Answer 1 : Thank you for this advice. We added the sentences as follows: “Pachymetry is here defined by the central corneal thickness and keratometry is defined by the corneal curvature of the two main corneal meridians.” And “SD-OCT gives two or three-dimensional images of the retina with near-cellular resolution, allowing ophthalmologists to analyze histologic-like images. SD-OCT uses near-infrared wavelength and so does not expose patients to radiations. The procedure for SFCT measurement that we used was previously described by Spaide et al. and is defined as the vertical distance from the hyperreflective line of Bruch’s membrane to the hyperreflective line of the inner surface of the sclera”

2/ You mentioned that IOP was measured using the Nidek Tonoref III, could you explain what is the assumed normal difference between measurements that is due to the variability of the device? because you found a significant change in IOP that I assume is considering that the variability between measures is 0, which probably is not true. Also, you say that the change in the IOP is 88% which considering that the initial mean is 16.35 and the 30 minutes post is 14.30 I am not sure if its right, I would say that a decrease of 2 mmHg if 16.35 is 100% represents 12.2% of decrease, would you agree on that? 

Answer 2 : Thank you for the notice: we added this sentence : “Nidek Tonoref III provides accurate and reliable measurements of IOP compared with Goldmann applanation tonometry. It is considered nowadays as the gold-standard for IOP measurements [7]. Even older devices of air tonometers show low variability ranging from -2mmHg to +2mmHg principally due to cardiac cycle, explaining why three measures should be taken in regular practice [8]. »

3/ In the 5.4. Intraocular Pressure point, you mention that no intraocular bubbles were identified in the eyes of the divers, how did you assess that? did you perform gonioscopy to assess properly the angle, or use anterior segment OCT? otherwise, it seems difficult to affirm that microscopic bubbles were not present at all, and that possibility might be highlighted or trying to seek another explanation.

Answer 3 : No gonioscopy was performed in our divers for two main reasons.  First for time management as we had already several examinations to perform in a short time laps. The second reason is that gonioscopy requires a contact lens and involves depression of the cornea. Such examination may transiently alter the cornea biometry and transparency adding variability to our measurements. Anterior segment OCT would have been a good examination but was not available during the sessions. We agree with the reviewer that anterior segment OCT examination would have been interesting in this study.

We are confident about the fact that potential bubbles trapped in trabecular meshwork did not interfere with our results. Such bubbles would block the trabecular meshwork and would increase IOP, which we do not find but rather the opposite in our results. If bubbles were indeed present in the trabecular meshwork, this presence was insufficient to increase IOP.

We deleted the sentence about bubbles since we agree with the reviewer, it doesn’t add much to the manuscript.

Why you did not consider adjusting the IOP with the corneal thickness? it is a shame as you had all the data collected to do it and it would add more accuracy to your results.

Answer 4 : Correction of IOP is of course important. It was not made initially in order to get the absolute data without any correction. We agree with the reviewer that it should have been done.

Though a correction would still be possible, it would not fundamentally change our results since pachymetry had low variability before and after the dive. In our results, pachymetry tends to slightly increase after the dive. Increased pachymetry can overestimate the IOP measurements. So we probably overestimated very slightly the IOP 30 minutes after the dive. This means the true IOP decrease should be even more important than what we described, but this change seemed neglectable to us.

Reviewer 2 Report

The aim of the article is to evaluate possible changes induced by scuba diving in ophthalmic parameters and especially subfoveal choroidal thickness (SFCT) and possible connection with central serous chorioretinopathy (CSR).

The study is original.

The English language is appropriate and understandable.

The manuscript is presented in a well structured manner.

References do not comprise self citations. Most of the references are not within last 5 years.

Images are appropriate and are easy to interpret.

In Introduction authors line 43, „ we were struck by the presence of common history of SCUBA diving among CSC patients” – this is very subjective observation, it will be useful if you can tell how many patients or percentage is suffering of CSC associated with SCUBA diving.

Line 44 “We know CSC is commonly associated with choroidal hyperpermeability and increased subfoveal choroidal thickness (SFCT), but its physiopathology is still poorly understood.” – can authors find some reference for this statement.

Methods:

Line 52 “Divers were otherwise healthy males between 28 and 72 years” – there are known differences in SFCT according to age, race, obesity etc. This group is very different according to age, then it is not stated are they Caucasian or other races. Also can it be clearly stated when the measures were taken according to diurnal variations of SFCT.

Line 81 “The same device was used for retinal autofluorescence imaging. Procedure for SFCT measurement has been previously described and is defined as the vertical distance from the hyperreflective line of Bruch’s membrane to the hyperreflective line of the inner surface of the sclera.” – reference for method of measurement is missing

Results

Line 162 – “Comparison (in percentage) of SFCT, IOP and Pachymetry between pre and post dive values at 30 minutes and 60 minutes post dive.(ns = not significant).” – results are show as percentages of values in table 1, while in table 2 are shown as real values, but without statistical p value.

The mean SFCT value in this paper is SFCT (µm) 327.1 (±102.0), so when we compare that value with values know in literature, Entezari et al. 363±84 µm in Iranian. Karapetyan et. Al Mean SFCT in Caucasians was 403.62 ± 37.4 µm, which are mentioned in paper. But there are also other authors in literature Zeried (Current medical imaging 2021) 285 ± 31 (range 203 to 399), or Ikuno Y (invest ophthalmol vis sci 2010) and Tan CS (invest ophthalmol vis sci 2012) from 191.5 to 342 µm, we can see that basic values are very different so the method is not precise.

Although the authors find statistical significance in SFCT the range of measurement and the variations of values within the group of 15 patients of very different age are doubtful for true comparison.

Discussion

Discussion is divided by different ophthalmic parameters and well written.

Author Response

The study is original.

The English language is appropriate and understandable.

The manuscript is presented in a well structured manner.

References do not comprise self citations. Most of the references are not within last 5 years.

Images are appropriate and are easy to interpret.

Thank you.

1/ In Introduction authors line 43, „ we were struck by the presence of common history of SCUBA diving among CSC patients” – this is very subjective observation, it will be useful if you can tell how many patients or percentage is suffering of CSC associated with SCUBA diving.

Answer 1 : In our department we noticed several patients suffering from central serous chorioretinopathy practicing SCUBA diving as recurrent hobby. We detected 6 cases of CSC in divers over about 150 CSC cases. That make about 4% of them. It seems not much but was sufficient to awake our interest as SCUBA diving is an uncommon activity and could be overlooked.

We agree with the reviewer and added a percentage according our clinical experience, we changed the sente as follows : „ we noticed the presence of common history of SCUBA diving among CSC patients representing around 4% of cases during a two-year period ”

Line 44 “We know CSC is commonly associated with choroidal hyperpermeability and increased subfoveal choroidal thickness (SFCT), but its physiopathology is still poorly understood.” – can authors find some reference for this statement.

  Answer 2 : We added the following references :

-           Semeraro F, Morescalchi F, Russo A, Gambicorti E, Pilotto A, Parmeggiani F, Bartollino S & Costagliola C. (2019). Central Serous Chorioretinopathy: Pathogenesis and Management. Clin Ophthalmol 13, 2341-2352.

-           Chung YR, Kim JW, Choi SY, Park SW, Kim JH & Lee K. (2018). SUBFOVEAL CHOROIDAL THICKNESS AND VASCULAR DIAMETER IN ACTIVE AND RESOLVED CENTRAL SEROUS CHORIORETINOPATHY. Retina 38, 102-107.

Methods:

3/ Line 52 “Divers were otherwise healthy males between 28 and 72 years” – there are known differences in SFCT according to age, race, obesity etc. This group is very different according to age, then it is not stated are they Caucasian or other races. Also can it be clearly stated when the measures were taken according to diurnal variations of SFCT.

Answer 3 : We agree with the reviewer. SFCT shows great variability between individuals. 130 µm, 300 µm 450 µm are all probable normal SFCT for different individuals. This is one of the reason that our results are expressed in percentage of pre-dive value. It doesn’t really matter if some had thinner or thicker SCFT than other. We only compared the variation in % of pre-dive value of the individual itself. Also SFCT tends to be maximal in the morning and decrease during the day. Both measurements were taken during a period of the day where SFCT is expected to decrease. Moreover, the dive duration was short and measurements were taken in the same period of the day.

We added that patients were all Caucasians and “Half of divers underwent measures during the morning and the other half in the afternoon. “ and “SFCT presents diurnal variation and tends to decrease over the day. Some divers underwent measures in the morning, other in the afternoon. Both are periods of decreasing SFCT so we don’t expect any difference between those two cohorts. Moreover, this circadian cycle does not explain the re-increase of SFCT 60 minutes after the dive. This points out an extra-physiological phenomenon allowing us to exclude this hypothesis.”

4/ Line 81 “The same device was used for retinal autofluorescence imaging. Procedure for SFCT measurement has been previously described and is defined as the vertical distance from the hyperreflective line of Bruch’s membrane to the hyperreflective line of the inner surface of the sclera.” – reference for method of measurement is missing

 Answer 4 : We corrected the sentence as follow: “The procedure for SFCT measurement that we used was previously described by Spaide et al. [9] and is defined as the vertical distance from the hyperreflective line of Bruch’s membrane to the hyperreflective line of the inner surface of the sclera”

Results

Line 162 – “Comparison (in percentage) of SFCT, IOP and Pachymetry between pre and post dive values at 30 minutes and 60 minutes post dive.(ns = not significant).” – results are show as percentages of values in table 1, while in table 2 are shown as real values, but without statistical p value.

 Answer 5 : We do understand that showing relative data and absolute data can be confusing. We think it is interesting to show the absolute data, yet not mandatory. But as authors, we focused mainly on the relative data and we invite our readers to do so. No conclusion can be done only by looking at the absolute data.

We decided to add P-values in table 2.

The mean SFCT value in this paper is SFCT (µm) 327.1 (±102.0), so when we compare that value with values know in literature, Entezari et al. 363±84 µm in Iranian. Karapetyan et. Al Mean SFCT in Caucasians was 403.62 ± 37.4 µm, which are mentioned in paper. But there are also other authors in literature Zeried (Current medical imaging 2021) 285 ± 31 (range 203 to 399), or Ikuno Y (invest ophthalmol vis sci 2010) and Tan CS (invest ophthalmol vis sci 2012) from 191.5 to 342 µm, we can see that basic values are very different so the method is not precise.

Although the authors find statistical significance in SFCT the range of measurement and the variations of values within the group of 15 patients of very different age are doubtful for true comparison.

Answer 6 : About the example chosen by the reviewer, the first one ranges from 203-399, which is very close to ours (327µm +- 102). The second one is indeed more different. Accordingly, authors themselves underline that SFCT is highly variable. They also develop their methods as follow “The choroidal thickness was measured using a line drawn perpendicularly from the hyperreflective line believed to represent the retinal pigment epithelium (RPE) to the choroid–scleral junction” which is the exact same as ours. Therefore we really tend to privilege the relative measurement. As each diver is his own control, basal SFCT does not interfere with the results. No matter if the initial SFCT is 100 or 500µm, it is the variation relative to the pre dive SFCT that matters. This variation expressed in % that we compare between individuals, not the raw data. But we agree with the reviewer that older divers may have increased stiffening of the vascular meshwork thus interfering with absolute results.

We changed accordingly : Literature presents sometimes normal SFCT ranges that may be different from ours [27] and SFCT seems to decrease over day time [9]. All these considerations show the importance of measuring variations taking the baseline measures as 100%, for each diver, allowing an appreciation of the magnitude of change rather than the absolute values.  

Discussion

Discussion is divided by different ophthalmic parameters and well written.

Round 2

Reviewer 2 Report

After reading authors letter and revised manuscript, my main comments were acknowledged. The manuscript was thoroughly revised according to the both reviewer comments. The quality of presentation is improved as the explanation of methods.

I do not have any other suggestions to the authors.

The final decision is now on main editor.

Good luck